# SMART-PC
# Skeletal Model Adaptation for Robust Test-Time Training in Point Clouds

**Ali Bahri** [1 2]  **Moslem Yazdanpanah** [1 2]  **Sahar Dastani** [1 2]  **Mehrdad Noori** [1 2]  **Gustavo Adolfo Vargas Hakim** [1 2]
**David Osowiechi** [1 2]  **Farzad Beizaee** [1 2]  **Ismail Ben Ayed** [1 2]  **Christian Desrosiers** [1 2]

## Abstract

Test-Time Training (TTT) has emerged as a promising solution to address distribution shifts in 3D point cloud classification. However, existing methods often rely on computationally expensive backpropagation during adaptation, limiting their applicability in real-world, time-sensitive scenarios. In this paper, we introduce SMART-PC, a skeleton-based framework that enhances resilience to corruptions by leveraging the geometric structure of 3D point clouds. During pre-training, our method predicts skeletal representations, enabling the model to extract robust and meaningful geometric features that are less sensitive to corruptions, thereby improving adaptability to test-time distribution shifts. Unlike prior approaches, SMART-PC achieves real-time adaptation by eliminating backpropagation and updating only BatchNorm statistics, resulting in a lightweight and efficient framework capable of achieving high frame-per-second rates while maintaining superior classification performance. Extensive experiments on benchmark datasets, including ModelNet40-C, ShapeNet-C, and ScanObjectNN-C, demonstrate that SMART-PC achieves state-of-the-art results, outperforming existing methods such as MATE in terms of both accuracy and computational efficiency. The implementation is available at: https://github.com/AliBahri94/SMART-PC.

## 1. Introduction

Recent advancements in deep learning have significantly improved the classification of 3D point clouds (Pang et al.,

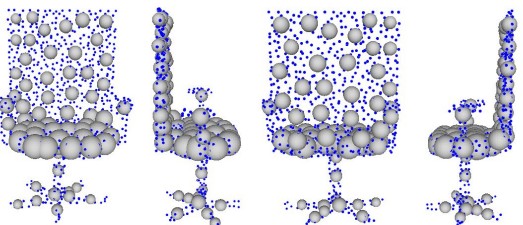

*Figure 1.* The blue points represent the sampled points on the surface of spheres created using the skeletal points as centers and their corresponding radii. Each sphere illustrates the local geometric structure defined by the skeletal representation.

2022; Zhang et al., 2022b; Bahri et al., 2025; Liang et al., 2024; Bahri et al., 2024a). However, these models often assume that the test data distribution matches the training distribution, an assumption that rarely holds in real-world scenarios. Distribution shifts, caused by factors such as environmental conditions or sensor inaccuracies in LiDAR data, can introduce distortions that degrade model performance (Ren et al., 2022; Sun et al., 2022). This vulnerability is particularly challenging in critical domains like autonomous navigation and robotics. As pre-training for all potential scenarios is impractical, robust methods for unsupervised test-time adaptation are essential to address these shifts effectively.

In Test-Time Training (TTT) for classification, an additional technique is employed during pre-training with the source dataset, which can be leveraged during the adaptation phase. During adaptation, the network utilizes unlabeled target data to dynamically adjust to shifts in data distributions at test time. Recent TTT approaches in the 2D image domain have explored various strategies, such as entropy-based regularization, updating BatchNorm statistics, and self-supervised tasks (Liang et al., 2020; Wang et al., 2020; Mirza et al., 2022; Sun et al., 2020). However, these methods struggle when directly applied to 3D point clouds, highlighting the need for TTT techniques specifically designed for 3D data. In the 3D domain, a notable work is the MATE framework (Mirza et al., 2023), which addresses the problem of TTT for 3D point cloud classification. MATE introduces a 3D-specific approach that leverages the self-supervised paradigm, where a network is adapted to out-of-distribution (OOD) target data by solving a self-supervised task. MATE

[1]LIVIA, ÉTS Montréal, Canada. [2]International Laboratory on Learning Systems (ILLS). Correspondence to: Ali Bahri <ali.bahri.1@ens.etsmtl.ca>.

utilizes a masked autoencoder (MAE), where the network reconstructs a point cloud from its partially removed data. This reconstruction process enables adaptation, enhancing robustness to distribution shifts during test-time.

While MATE represents a significant advancement in TTT for 3D point clouds, it has critical limitations. MATE focuses on reconstructing points on the surface of objects, making it highly sensitive to surface-level corruptions, such as noise, which frequently occur in real-world scenarios. This focus on surface points makes the features learned by MATE highly sensitive to corruptions, resulting in reduced resilience to distribution shifts. Additionally, MATE heavily depends on updating all network parameters during adaptation. This approach reduces the model's speed, making it less practical for real-time applications due to its lower Frames/Second.

In contrast, our method overcomes these limitations by introducing a skeleton-based framework for TTT. By capturing the essential geometric structure of 3D point clouds, skeletons enable the model to learn powerful and meaningful features that are inherently less sensitive to corruptions. This is achieved through the abstraction property of skeletal representations (Lin et al., 2021), which simplifies the shape into a compact form, filtering out high-frequency noise and local distortions. As a result, the model focuses on the underlying geometric structure rather than surface-level variations, enhancing robustness to distribution shifts. By leveraging these robust features, our approach eliminates the need for backpropagation during adaptation, significantly improving speed. Instead, following prior studies (Nado et al., 2020; Li et al., 2018), we update only the BatchNorm statistics, which effectively adapts the model to distribution shifts without modifying its learned weights. This efficiency demonstrates that skeleton-based pre-training inherently equips the model with features that are more resilient to corruption compared to MATE, providing a lightweight and effective solution for TTT. As shown in Figure 1, the skeletal representation captures the geometric structure of point clouds by abstracting them into compact and meaningful features. Our method not only overcomes the limitations of MATE but also introduces a paradigm shift in TTT by emphasizing the importance of meaningful feature extraction during pre-training, ensuring robustness and adaptability with minimal computational effort. Our main contributions are:

- We propose SMART-PC, a skeleton-based framework for TTT that enhances robustness by extracting geometric features that are less sensitive to distribution shifts.
- Our method achieves test-time adaptation without backpropagation, relying solely on updating BatchNorm statistics, making it highly efficient for real-time applications.

- We achieve state-of-the-art results on benchmark datasets, demonstrating the effectiveness of our approach compared to existing methods, including MATE.

## 2. Related Works

**Test-Time Training.** TTT enhances model adaptability by leveraging unseen target data during inference, unlike domain generalization or adaptation methods, which operate solely during training. TTT methods can be broadly categorized into regularization-based and self-supervised approaches. Regularization-based methods include TENT (Wang et al., 2020), which minimizes prediction entropy by adapting BatchNorm parameters, and SHOT (Liang et al., 2020), which combines entropy minimization with diversity regularization to develop robust feature extraction. MEMO (Zhang et al., 2022a) applies data augmentations to test inputs and minimizes the entropy of averaged outputs, improving robustness. DUA (Mirza et al., 2022) updates BatchNorm statistics to address distribution shifts with minimal overhead, while T3A (Iwasawa & Matsuo, 2021) refines the linear classifier using pseudo-prototypes, achieving backpropagation-free adaptation. These methods highlight the diversity of TTT strategies for addressing distribution shifts.

Among self-supervised approaches, TTT++ (Liu et al., 2021) introduces an additional self-supervised branch that leverages contrastive learning within the source model to facilitate adaptation to the target domain. Another method, TTT-MAE (Gandelsman et al., 2022), employs a Masked Autoencoder (MAE) to address the one-sample learning problem effectively, demonstrating improved performance across various visual benchmarks. A recent method, MATE (Mirza et al., 2023), is the first TTT framework specifically tailored for 3D point cloud data, enhancing the resilience of deep networks to distribution shifts during testing. It leverages a MAE objective, masking a significant portion of each target point cloud and tasking the network with reconstructing the complete structure before classification. In addition to these approaches, several works on test-time adaptation have explored updating model parameters during inference to handle distribution shifts effectively (Wang et al., 2024; Bahri et al., 2024b).

**Skeletal Representation.** Skeletal representations are a compact and structural abstraction of 3D point clouds, capturing the underlying geometric and topological properties of objects. These representations simplify complex point cloud data by reducing it to a set of key skeletal points that capture the core of the geometry (Figure 1), which are particularly useful for tasks such as shape analysis, object recognition, and reconstruction. Several methods have been proposed to generate skeletal representations from 3D data.

Early approaches often relied on medial axis transformations (MAT) (Choi et al., 1997) or Voronoi-based (Ogniewicz & Ilg, 1992) methods to extract the central skeleton of an object. While effective in capturing global geometry, these methods are typically computationally intensive and sensitive to noise (Li et al., 2015; Sun et al., 2015; Yan et al., 2018). More recent learning-based approaches, such as Point2Skeleton (Lin et al., 2021) and Learnable Skeleton-Aware 3D Point Cloud Sampling (Wen et al., 2023), have shifted towards using neural networks to predict skeletal points directly from input point clouds. These methods leverage local geometric features and end-to-end optimization to produce accurate and robust skeletal representations. By capturing the core geometric structure of 3D point clouds, skeletal representations provide a compact and meaningful abstraction that is less sensitive to noise and distortions compared to raw point cloud data. This robustness makes them particularly suitable for dynamic adaptation scenarios, where models need to generalize effectively to unseen conditions. In this paper, we leverage skeletal representations with technical enhancements that enable the network to learn robust geometric features, thereby eliminating the need for extensive parameter updates during Test-Time Adaptation (TTA). Instead, our method relies solely on lightweight adjustments, such as updating BatchNorm statistics. This approach ensures both efficient and effective TTT while fully utilizing the inherent strengths of skeletal representations.

## 3. Method

### 3.1. Preliminaries

A point cloud is represented as $P \in \mathbb{R}^{N \times 3}$, where $N$ is the total number of points, and each point is defined by its 3D spatial coordinates $(x, y, z)$. This representation captures the geometric structure of 3D objects, but due to the irregular and unordered nature of point clouds, processing all $N$ points can be computationally expensive and redundant. To reduce redundancy and focus on the most representative points, we apply Farthest Point Sampling (FPS). This technique selects a subset of $M$ points from the original point cloud as *centers*, denoted as:

$$C = FPS(P) \in \mathbb{R}^{M \times 3}, \tag{1}$$

where $M \ll N$. The FPS algorithm ensures that the selected centers are evenly distributed across the point cloud, preserving the overall geometric structure. For each center point $c_i \in C$, we construct its local neighborhood by selecting its $K$ nearest neighbors from the original point cloud $P$ using the K-Nearest Neighbors (KNN) algorithm. This results in a neighborhood tensor:

$$P_{local} = kNN(C, P) \in \mathbb{R}^{M \times K \times 3}, \tag{2}$$

where $K$ is the number of neighbors for each center point. Each neighborhood $P_{local}[i] \in \mathbb{R}^{K \times 3}$, corresponding to a

center $c_i$, captures the local geometric features around $c_i$. These neighborhoods provide localized context, allowing the network to efficiently process the point cloud by focusing on both global structure (via the centers) and local geometric details (via the neighbors). This hierarchical tokenization forms the basis for subsequent feature extraction and skeletal prediction.

With the point cloud tokenized into centers and their local neighborhoods, the next step is to define the skeletal representation. The skeleton of a 3D shape provides a compact representation of its intrinsic geometric structure. It abstracts the shape by representing it as a collection of skeletal points and their corresponding radii, which together form a *skeletal mesh*. This concept was introduced by (Lin et al., 2021) for learning-based approaches. A skeletal mesh is defined as a discrete set of skeletal spheres. Each sphere is represented as:

$$s = \left( c_s, r(c_s) \right) \in \mathbb{R}^4, \tag{3}$$

where $c_s \in \mathbb{R}^3$ is the center of the skeletal sphere, referred to as the *skeletal point*, and $r(c_s) \in \mathbb{R}$ is its associated radius, representing the distance from the center $c_s$ to the surface of the original 3D shape. The skeletal mesh is constructed by connecting the skeletal spheres using two main elements. First, *edges*, denoted as $e_{ij} = (c_i, c_j)$, connect two skeletal points $c_i$ and $c_j$, forming the basic linear structure of the skeleton. Second, *faces*, denoted as $f_{ijk} = (c_{s_i}, c_{s_j}, c_{s_k})$, are triangular connections between three skeletal points that capture 2D non-manifold surfaces within the mesh.

The skeletal representation possesses two key properties. The first is **recoverability**, where the skeletal mesh acts as a complete descriptor of the shape. This means the original 3D shape can be reconstructed by interpolating the skeletal spheres. For a skeletal sphere $s = (c_s, r(c_s))$, any point $p$ on its surface can be expressed as:

$$p = c_s + r(c_s) \cdot v, \tag{4}$$

where $v$ is a unit vector specifying the direction. The second property is **abstraction**, by which the skeleton simplifies the representation of the shape by capturing its fundamental structure, significantly reducing the complexity of the original point cloud while retaining its essential geometric information. This discrete skeletal representation approximates the Medial Axis Transform (MAT), which is a continuous formulation of the skeleton. Unlike the MAT, the skeletal mesh is more robust to boundary noise and perturbations, making it suitable for learning-based approaches. By predicting skeletal points and their radii, the skeletal representation provides a robust and efficient way to analyze and process 3D shapes, even under noisy or corrupted conditions.

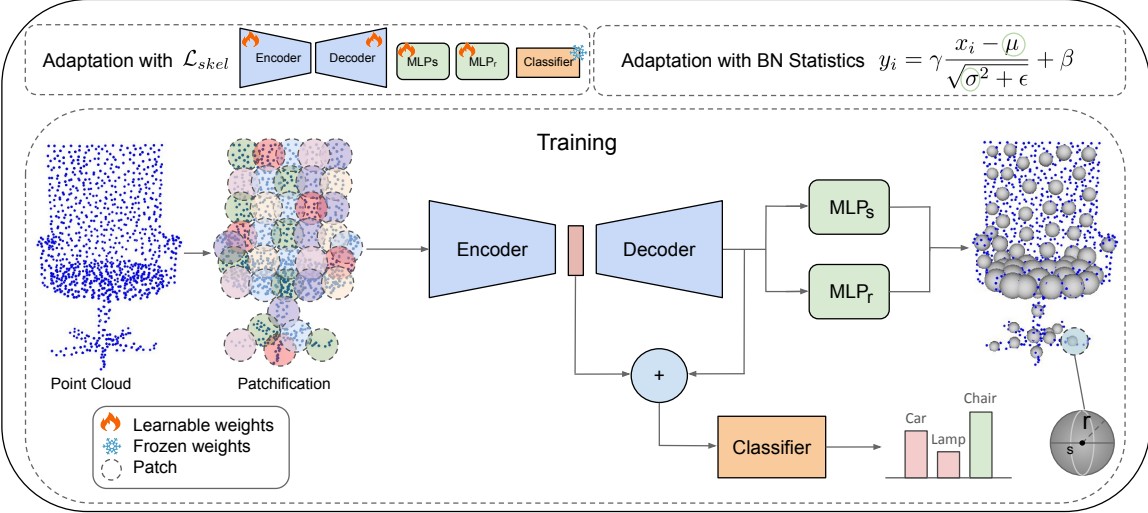

*Figure 2.* An overview of the SMART-PC framework. The framework integrates skeletal prediction and classification tasks, leveraging skeletal representations to extract robust geometric features. During online adaptation, two strategies are employed: adaptation with the skeletal loss $\mathcal{L}_{skel}$ and backpropagation, and a lightweight, backpropagation-free approach that updates only BatchNorm statistics.

### 3.2. Overview

In this paper, we propose a novel skeleton-based framework for Test-Time Training (TTT) designed to improve the robustness and adaptability of 3D point cloud classification under distribution shifts. Our method consists of three components: *Framework*, *Training*, and *Test-Time Adaptation* (TTA).

In the *Framework* component, we define the structure of the model, which includes components for both classification and skeletal point prediction. This involves investigating the architectural details and ensuring the network is capable of handling both tasks effectively. During the *Training* phase, the model is trained on source data to perform two tasks: skeletal prediction and classification. The skeletal-based pre-training enables the model to learn robust and meaningful geometric features that enhance resistance to corruptions and generalize effectively across varying distributions. In the TTA phase, our method eliminates the need for backpropagation when the adaptation strategy is online, meaning the model is not reset after each epoch. Instead, it relies solely on lightweight updates to the BatchNorm statistics, enabling the model to adapt dynamically to target data while achieving a higher frame rate for real-time applications. The overall network architecture is illustrated in Figure 2.

### 3.3. Framework

Our framework is designed to perform two primary tasks: skeletal point and radius prediction, and point cloud classification. These tasks share a unified encoder while leveraging separate network components for their specific objectives.

The framework is divided into two key branches: the *skeletal branch* for predicting skeletal representations and the *classification branch* for predicting class labels.

**Skeletal Branch.** The skeletal branch predicts skeletal points $S \in \mathbb{R}^{M \times 3}$ and their corresponding radii $R \in \mathbb{R}^{M \times 1}$ from the input point cloud. The encoder, denoted as $E$, processes the tokenized point cloud $P_{local} \in \mathbb{R}^{M \times K \times 3}$ to extract global and local geometric features:

$$F_{enc} = E(P_{local}) \in \mathbb{R}^{M \times d}, \quad (5)$$

where $d$ is the dimensionality of the feature space. The encoder captures both local geometric details and global structural information.

The decoder, denoted as $D$, refines the encoded features to produce a contextually enriched representation:

$$F_{dec} = D(F_{enc}) \in \mathbb{R}^{M \times d}. \quad (6)$$

To predict skeletal points and radii, two separate Multi-Layer Perceptrons ($MLP_s$ and $MLP_r$) are applied to the refined features $F_{dec}$. The skeletal points are predicted as:

$$c_s = MLP_s(F_{dec}) \in \mathbb{R}^{M \times 3}, \quad (7)$$

and the radii are predicted as:

$$r = MLP_r(F_{dec}) \in \mathbb{R}^{M \times 1}. \quad (8)$$

The skeletal branch leverages these predictions to model the underlying geometric structure of the input point cloud. Unlike existing skeletal prediction methods (Lin et al., 2021;

Wen et al., 2023), which rely on convex combinations of input points to generate skeletal points, our approach directly predicts the skeletal points and radii from the refined features $F_{dec}$. Convex combination methods inherently inject raw point cloud data into the prediction process, making the task easier for the network. However, this dependency on the raw point cloud can hinder the encoder's ability to learn robust and meaningful geometric features, especially in the presence of noise or distribution shifts.

**Classification Branch.** The classification branch uses the shared encoder to extract features for class prediction. To enhance the classification performance, the features from the encoder $F_{enc}$ and decoder $F_{dec}$ are combined by a sum:

$$F_{combined} = F_{enc} + F_{dec}. \qquad (9)$$

This combination allows the classification head to leverage high-level features from the decoder that are closely related to the skeletal points, alongside the global context from the encoder. By incorporating these skeletal-related features, the model enhances its ability to classify point clouds, particularly by utilizing structural information that aids in distinguishing complex or similar classes. The combined features are passed through the classification head, which consists of multiple MLP layers, normalization, and dropout layers, to predict class probabilities:

$$p = Softmax(MLP_{cls}(F_{combined})) \in \mathbb{R}^K, \qquad (10)$$

where $K$ is the number of classes.

### 3.4. Training

In our framework, the skeletal branch is trained in a self-supervised manner since no labels are available for skeletal points, while the classification branch is trained in a supervised manner using labeled source data. To optimize these branches, we adopt loss functions inspired by prior works in skeletal representation (Lin et al., 2021; Wen et al., 2023). Below, we detail the loss functions used for training the skeletal and classification branches.

**Skeletal Losses.** The skeletal branch optimizes three complementary loss functions to ensure accurate prediction of skeletal points and their corresponding radii. Unlike existing skeletal methods that operate at the point level, our approach predicts skeletal points and radii at the patch level. For each patch, the model predicts the center of a skeletal sphere and its radius, which together abstract the local structure of the point cloud.

**1) Point-to-Sphere Loss.** This loss ensures that input points lie on the surface of their corresponding skeletal spheres and that skeletal spheres align closely with their associated input points $P$:

$$\mathcal{L}_{p2s} = \sum_{p \in P} \left( \min_{s \in S} \|p - c_s\|_2 - r(c_s) \right)$$
$$+ \sum_{s \in S} \left( \min_{p \in P} \|c_s - p\|_2 - r(c_s) \right), \qquad (11)$$

where $c_s$ and $r(c_s)$ again denote the center and radius of skeletal sphere $s$.

**2) Sampling Loss.** To further ensure alignment between the skeletal spheres and the input point cloud, the Chamfer Distance between sampled points on the surface of skeletal spheres and the input points is calculated:

$$\mathcal{L}_{sampling} = \sum_{p \in P} \min_{t \in T} \|p - t\|_2 + \sum_{t \in T} \min_{p \in P} \|t - p\|_2, \quad (12)$$

where $T$ represents points sampled uniformly on the surfaces of the skeletal spheres. This loss aggregates geometric information from the neighborhood of input points, effectively reducing high-frequency noise. As each skeletal point represents the local center of a region in $P$, the influence of individual noisy points is averaged, resulting in $s$ being less sensitive to corruption.

**3) Radius Regularization Loss.** To avoid the instability caused by overly small radii due to noise, a regularization term that encourages larger radii is also used:

$$\mathcal{L}_{radius} = -\sum_{s \in S} r(c_s). \qquad (13)$$

Furthermore, the radii $r$, predicted alongside $c_s$, capture the extent of each skeletal point's influence. This loss encourages larger radii, reducing sensitivity to localized noise by ensuring that each skeletal sphere represents a broader region of the point cloud. This abstraction minimizes the effect of outliers, further enhancing robustness.

The total skeletal loss is a weighted combination of these three terms:

$$\mathcal{L}_{skel} = \mathcal{L}_{p2s} + \lambda_1 \mathcal{L}_{sampling} + \lambda_2 \mathcal{L}_{radius}, \qquad (14)$$

where $\lambda_1 \geq 0$ and $\lambda_2 \geq 0$ are hyperparameters balancing the contributions of each loss.

**Classification Loss.** For the classification branch, we train the network using labeled source data. The features from the encoder and decoder are combined as described earlier and passed to the classification head. The classification loss is defined as the cross-entropy between the predicted probabilities and the ground truth labels:

$$\mathcal{L}_{cls} = -\frac{1}{B} \sum_{i=1}^{B} \sum_{k=1}^{K} y_{ik} \log(\hat{y}_{ik}), \qquad (15)$$

where $B$ is the batch size, $K$ is the number of classes, $y_{ik}$ is the ground truth label for the $i$-th sample in class $k$, and $\hat{y}_{ik}$ is the predicted probability for the same class.

The overall loss for training the network is a combination of the skeletal and classification losses:

$$\mathcal{L}_{total} = \mathcal{L}_{skel} + \mathcal{L}_{cls}. \quad (16)$$

This joint optimization allows the model to learn robust skeletal representations in a self-supervised manner while simultaneously leveraging labeled source data for classification.

### 3.5. Test-Time Adaptation (TTA)

In this work, we explore two modes of test-time adaptation, inspired by prior works such as MATE (Mirza et al., 2023): *online adaptation* and *standard adaptation*. Both modes aim to adapt the model to corrupted target data during test time but differ in how the model state is managed across epochs and corruptions.

**Online Adaptation.** In the online mode, the model is not reset after each epoch, allowing the accumulated information from previous batches to influence the adaptation process. However, the model is reset at the beginning of adaptation for each new corruption type. Initially, we perform test-time adaptation in a backpropagation-free manner, updating only the statistical parameters of the Batch Normalization (Batch-Norm) layers (i.e., the running mean and running variance). This approach leverages the observation that the features extracted by the model during pre-training on source data are highly robust and less sensitive to corruption. By updating only the BatchNorm statistics, the model dynamically adjusts to the target data, achieving higher frame rates for real-time applications while maintaining high performance.

To further analyze the effectiveness of the extracted features, we conducted an additional experiment where all model parameters, along with the BatchNorm statistical parameters, were updated during test-time adaptation. Specifically, we optimized the skeletal loss functions $\mathcal{L}_{p2s}$, $\mathcal{L}_{sampling}$, and $\mathcal{L}_{radius}$ (as described in Section 3.4) to adapt the skeletal representation to the target distribution.

**Standard Adaptation.** In the standard mode, the model is reset after every batch, meaning each test batch adapts independently without accumulating information from previous batches. In this case, updating only the BatchNorm statistics has limited effectiveness since the adaptation does not retain context across batches. Consequently, for standard adaptation, we update all the parameters of the model using the skeletal loss functions $\mathcal{L}_{p2s}$, $\mathcal{L}_{sampling}$, and $\mathcal{L}_{radius}$. This allows the model to adjust more comprehensively to each batch of corrupted data.

## 4. Experiments

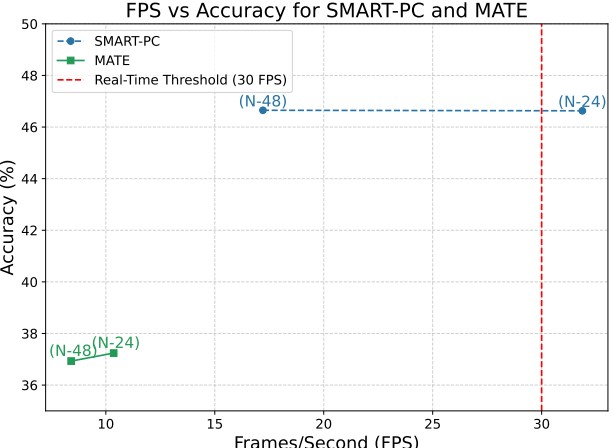

*Figure 3.* Frames/Second vs. Accuracy for SMART-PC and MATE during online adaptation on the ScanObjectNN dataset. SMART-PC achieves higher frame rates and accuracy, surpassing the real-time threshold (30 Frames/Second) in the (N-24) setting.

In this section, we present a comprehensive evaluation of our proposed method using multiple 3D point cloud datasets. To thoroughly examine its robustness and generalization capabilities, we conduct experiments on three benchmark datasets: ModelNet40-C (Sun et al., 2022), ShapeNet-C (Chang et al., 2015), and ScanObjectNN-C (Uy et al., 2019). These datasets encompass a variety of real-world challenges, including different levels of corruption and noise, enabling us to showcase the effectiveness of our approach in diverse and complex scenarios. Furthermore, we evaluate the performance of our method in both online and standard modes. For the online mode, we compare the backpropagation-free strategy with backpropagation-based adaptation to highlight the efficiency and robustness of our approach. A detailed description of the dataset, additional experiments, expanded visualization results related to skeletons, and a comprehensive evaluation of our method across all datasets, including accuracy metrics for each corruption type, are provided in the Supplementary Materials.

### 4.1. Implementation Details

For both training and adaptation, we employed the Point-MAE backbone, following the settings outlined in the MATE paper (Mirza et al., 2023) to ensure a fair and consistent comparison. The batch size is set to 1 for both adaptation modes, with 1 iteration for the online mode and 20 iterations for the standard mode, identical to the MATE paper for fair comparison once again. The optimizer and learning rate are identical to those used in the MATE paper. For augmentation, *scale-transfer* is used during pre-training, similar to MATE. During adaptation, however, MATE em-

*Table 1.* Top-1 Classification Accuracy (%) for the ModelNet-40C, ScanObjectNN-C and ShapeNet-C datasets. Differences between SMART-PC variants and MATE counterparts are indicated as (± x%). * denotes reproduced results, † denotes adaptation without backpropagation, and ‡ denotes diffusion-based test-time adaptation methods.

| | ModelNet-40C | ScanObjectNN-C | ShapeNet-C |
|---|---|---|---|
| Org-SO* | 54.0 | 37.0 | 61.3 |
| MATE-SO* | 54.4 | 34.5 | 56.5 |
| SMART-PC-SO | **61.7** (+7.3%) | **38.7** (+4.2%) | **64.5** (+8.0%) |
| DUA (Mirza et al., 2022) | 54.7 | 38.7 | 60.8 |
| TTT-Rot (Sun et al., 2020) | 53.0 | – | 60.9 |
| SHOT (Liang et al., 2020) | 26.6 | 29.6 | 36.2 |
| T3A (Iwasawa & Matsuo, 2021) | 55.7 | – | 54.4 |
| TENT (Wang et al., 2020) | 26.5 | 27.7 | 37.4 |
| CloudFixer-Standard‡ (Shim et al., 2024) | 68.0 | - | - |
| CloudFixer-Online‡ (Shim et al., 2024) | 77.2 | - | - |
| DDA-Standard‡ (Gao et al., 2023) | 68.1 | - | - |
| SVWA-Standard* (Bahri et al., 2024b) | 57.1 | 37.4 | 50.5 |
| MATE-Standard* (Mirza et al., 2023) | 63.0 | 36.9 | 63.1 |
| SMART-PC-Standard | **63.1** (+0.1%) | **39.6** (+2.7%) | **64.4** (+1.3%) |
| BFTT3D* (Wang et al., 2024) | 57.2 | 33.0 | 60.7 |
| MATE-Online* (Mirza et al., 2023) | 70.6 | 36.9 | **69.1** |
| SMART-PC-Online† | 70.8 | 46.7 | 65.9 |
| SMART-PC-Online | **72.9** (+2.3%) | **47.4** (+10.5%) | 67.1 (-2.0%) |

ploys random masking, whereas we use random rotation in all experiments. All experiments were conducted using a single NVIDIA A6000 GPU, ensuring consistency across all tested configurations. Additionally, we evaluate the frames per second rate of our method and compare it with MATE.

**Real-Time Adaptation.** Figure 3 illustrates the adaptation performance of SMART-PC and MATE in terms of Frames/Second on the ScanObjectNN-C dataset. The adaptation strategy employed is online, with a batch size of 1 and a single iteration per adaptation step. The terms "N-48" and "N-24" represent the number of created batches, which are generated differently for the two methods: MATE uses random masking, while SMART-PC employs random rotations.

A significant advantage of SMART-PC is its ability to perform adaptation in online mode without requiring backpropagation. By updating only the statistics of BatchNorm layers, SMART-PC achieves significantly higher Frames/Second compared to MATE, which relies on computationally expensive backpropagation during adaptation. As shown in Figure 3, reducing the number of batches from N-48 to N-24 does not affect the accuracy of either method. However, MATE cannot increase its Frames/Second rate due to the inherent limitations of backpropagation. In contrast, SMART-PC demonstrates a substantial increase in Frames/Second, surpassing the real-time threshold of 30 Frames/Second when fewer batches are used. This highlights the efficiency and scalability of SMART-PC, making it a practical solution for real-world applications requiring real-time adaptation.

For additional comparisons with other methods, including diffusion-based approaches, please refer to the Supplementary Materials.

## 4.2. Main Results

In all results tables, "Org-SO" refers to the evaluation of a naive pretrained model, without any additional pretraining branch, on the corrupted dataset without adaptation. "MATE-SO" represents a model pretrained with a reconstruction branch, evaluated on the corrupted dataset without adaptation. "SMART-PC-SO" denotes our model pretrained with a skeleton branch on clean data, evaluated on the corrupted dataset without adaptation. Results marked with * indicate reproduced outcomes, and "†" refers to adaptation using BatchNorm statistical parameters without backpropagation.

**ModelNet-40C.** Table 1 presents the top-1 classification accuracy on the ModelNet40-C dataset under various corruption types. In the source-only setting, SMART-PC-SO achieves an average accuracy of 61.7%, significantly outperforming both MATE-SO (53.7%) and Org-SO (54.0%). This demonstrates the robustness of our skeleton-based pretraining in handling distribution shifts without adaptation.

For adaptation strategies, in the standard mode, SMART-PC-Standard achieves an average accuracy of 63.1%, higher than MATE-Standard (63.0%). In the online mode, SMART-PC-Online attains the highest average accuracy of 72.9%, compared to 69.6% for MATE-Online. Notably, SMART-PC-Online† leverages a backpropagation-free strat-

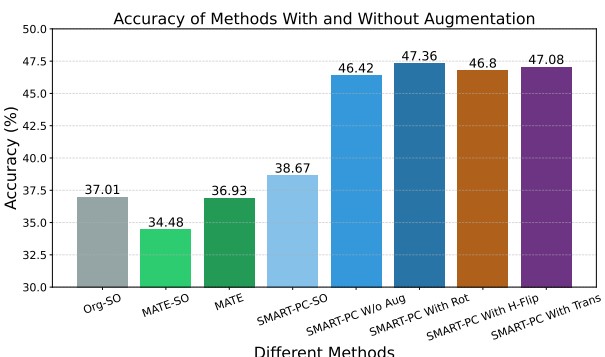

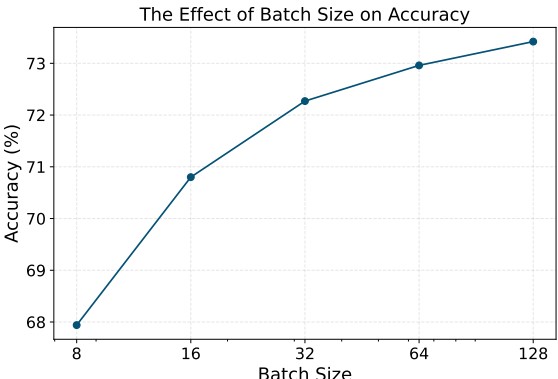

*Figure 4.* Effect of Augmentation During Adaptation in Our Method and Comparison with Other Methods on the ScanObjectNN Dataset in Online Mode.

*Figure 5.* Impact of Batch Size on Mean Accuracy for the ModelNet-C Dataset in Standard Mode.

egy, which enables efficient adaptation while maintaining superior performance. These results highlight the effectiveness and scalability of SMART-PC in both source-only (without adaptation) and adaptive settings under challenging conditions.

**ShapeNet-C.** Table 1 summarizes the top-1 classification accuracy on the ShapeNet-C dataset under various corruption types. In the source-only setting, SMART-PC-SO achieves an average accuracy of 64.5%, outperforming both MATE-SO (56.5%) and Org-SO (61.3%). This result highlights the effectiveness of our skeleton-based pre-training in generalizing to unseen corruptions without adaptation.

**ScanObjectNN-C.** Table 1 presents the top-1 classification accuracy across 15 corruption types on the ScanObjectNN-C dataset. In the source-only setting, SMART-PC-SO achieves an average accuracy of 38.7%, significantly outperforming both MATE-SO (34.5%) and Org-SO (37.0%).

In the standard adaptation mode, SMART-PC-Standard achieves an average accuracy of 39.6%, slightly higher than MATE-Standard (36.9%). This improvement highlights the effectiveness of leveraging skeletal representations for enhancing robustness to distribution shifts.

In the online mode, SMART-PC-Online achieves the highest average accuracy of 47.4%, significantly outperforming MATE-Online (36.9%). Furthermore, SMART-PC†, which utilizes a backpropagation-free adaptation strategy by updating only the BatchNorm statistics, achieves a competitive accuracy of 46.7%. The efficiency of the backpropagation-free strategy highlights its practicality for real-world applications requiring high Frames per Second.

Overall, these results demonstrate that SMART-PC consistently achieves state-of-the-art performance across all adaptation modes, with notable gains under challenging real-world corruptions, highlighting its robustness and generalization capability.

## 4.3. Ablation Study

**Batch Size.** We evaluate the effect of batch size on our method's performance using the ModelNet40-C dataset in standard adaptation mode, where all parameters are updated, with 20 iterations as in the MATE paper. As shown in Figure 5, increasing the batch size improves accuracy, from 67.94% at batch size 8 to 73.42% at batch size 128. This improvement showcases the effectiveness of larger batch sizes for achieving higher accuracy. To ensure fairness in comparison with MATE and other methods and to avoid increasing computational costs, we used a batch size of 1 for all main results in both standard and online modes.

**Augmentation.** Figure 4 illustrates the impact of different augmentations and the absence of augmentation on our method during adaptation. This experiment was conducted on the ScanObjectNN dataset in online mode with batch size 1 and iteration 1. Consistent with the MATE paper, we created 48 batches using random rotations to introduce diversity among the data. Even without augmentation (*SMART-PC W/O Aug*), our method achieves a 46.42% accuracy, outperforming *MATE* (36.93%) and demonstrating the robustness of our skeleton-based framework. Applying random rotations (*SMART-PC With Rot*) increases accuracy to 47.36%. Other augmentations, including horizontal flipping (*SMART-PC With H-Flip*) and translations (*SMART-PC With Trans*), achieve comparable improvements, with accuracies of 46.8% and 47.08%, respectively.

**Feature Summation.** We conduct an ablation study on the impact of summing the features from the shared encoder, used for both skeletal and classification tasks, with the decoder features. This summation improves SMART-PC's accuracy from 62.4% to 63.1% on the ModelNet-C dataset in standard mode, compared to using only the encoder features for classification.

**Skeleton Loss Coefficients.** To further evaluate the contribution of each loss component in our skeleton-based pre-

| Pt2Sphere | Sampling | RadiusReg | Source Acc(%) | Corrupted Acc(%) |
|-----------|----------|-----------|---------------|------------------|
| 1.0 | 1.0 | 0.0 | 91.3 | 67.82 |
| 0.0 | 1.0 | 1.0 | 91.6 | 67.79 |
| 1.0 | 0.0 | 1.0 | 91.6 | 67.80 |
| 1.0 | 1.0 | 1.0 | 91.2 | 72.84 |
| **0.3** | **1.0** | **0.4** | **91.3** | **72.95** |

*Table 2.* Ablation study of skeleton loss coefficients on ModelNet40 and ModelNet40-C (online adaptation). The best configuration corresponds to the original coefficients from the Point2Skeleton paper.

| Dataset | Org-SO | MATE-SO | SMART-PC-SO |
|---------|--------|---------|-------------|
| ScanObjectNN-C | 33.00 | 33.22 | **35.90** |
| ModelNet40-C | 57.16 | 54.71 | **65.25** |
| ShapeNet-C | 60.73 | 53.07 | **62.24** |

*Table 3.* Mean accuracy (%) of BFTT3D using different pretrained models under the backpropagation-free setting. SMART-PC-SO achieves the best results across all datasets.

training, we conducted an ablation study using different coefficients for the Skeletal loss terms: point-to-sphere loss, sampling loss, and radius regularization. The experiments were performed on the ModelNet40 and ModelNet40-C datasets under the online adaptation setting.

As shown in Table 2, the best performance is obtained with the coefficient set $(0.3, 1.0, 0.4)$, which corresponds to the original settings in the Point2Skeleton paper (Lin et al., 2021). This configuration achieves the highest accuracy of 72.95% on corrupted data (Corrupted Acc), confirming that each skeletal loss term contributes meaningfully to the learning of robust features under corruption. Additionally, as described in the main paper, the Radius Regularization Loss (Equation 6) is designed to avoid instability caused by overly small radii, especially under noisy conditions. This loss encourages the model to learn larger and more stable radii, which improves the robustness of the skeletal abstraction. Although we do not observe excessively large radii, the Point-to-Sphere and Sampling losses (Equations 11 and 12) implicitly constrain radius size by preserving geometric consistency. As shown in Table 2, removing the regularization term leads to a drop in performance, confirming its importance.

**Evaluating Pretraining Strategies in Backpropagation-Free Adaptation.** Our method supports two adaptation modes: one with backpropagation and one that is backpropagation-free. The goal of the backpropagation-free mode is to show that pretraining with a skeleton-based decoder encourages the model to learn robust and meaningful geometric features. These features are resilient enough that, during test-time, simply updating the BatchNorm statistical parameters (i.e., running mean and variance) is sufficient to improve performance—without performing gradient-based updates.

To validate this effect, we conducted additional experiments

using the BFTT3D (Wang et al., 2024) adaptation method across three different pretraining strategies (Org-SO, MATE-SO, and SMART-PC-SO). Each pretrained model was evaluated using the same BFTT3D adaptation strategy under the backpropagation-free setting. As shown in Table 3, our SMART-PC-SO model consistently outperforms both Org-SO and MATE-SO across all three datasets. This provides strong evidence that the skeletal decoder encourages the model to extract more structure-aware and corruption-resilient features, which support effective test-time adaptation without updating model weights.

## 5. Conclusion

In this paper, we proposed SMART-PC, a skeleton-based framework for robust and efficient test-time training of 3D point cloud models under distribution shifts. By leveraging skeletal representations, SMART-PC enhances robustness to corruptions while enabling high Frames per Second during online adaptation through a backpropagation-free strategy that updates only BatchNorm statistics. Experiments on benchmark datasets, including ModelNet40-C, ShapeNet-C, and ScanObjectNN-C, demonstrate state-of-the-art performance in both source-only and adaptation settings. Overall, SMART-PC provides a scalable and practical solution for real-world applications requiring robust 3D point cloud classification under challenging conditions. Future work may explore extending this framework to other 3D tasks, such as semantic segmentation, instance segmentation, and object detection.

## Impact Statement

This paper presents work aimed at advancing the field of Machine Learning. While there are several potential societal implications of our work, we believe that none of them need to be specifically highlighted in this context.

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

# A. Implementation

Our approach was implemented using PyTorch, with the codebase organized into two main components: *Pretrain* and *Adaptation*.

**Pretrain.**   We start with the initial pretraining phase of the base models (Point-MAE). In this phase, the backbone is pretrained with a classification branch in a fully supervised manner and a skeleton branch in a self-supervised manner. The pretraining is performed on clean datasets such as ModelNet, ShapeNet, and ScanObjectNN, ensuring the models are well-prepared for the subsequent adaptation steps.

**Adaptation.**   After completing the pretraining phase, we transition to the adaptation stage, which consists of two modes: **online** and **standard**. In the **standard adaptation mode**, all model parameters are updated using the skeleton loss, allowing the model to comprehensively adjust to the target data. In the **online adaptation mode**, we employ two distinct strategies. The first strategy involves adapting only the statistical parameters of the BatchNorm layers (e.g., running mean and variance) without backpropagation. This approach significantly reduces computational costs, enabling our method to achieve higher Frames per Second and making it suitable for real-time applications. The second strategy involves updating all model parameters using the skeleton loss, similar to the standard mode. These flexible adaptation strategies highlight the efficiency and scalability of our method, catering to both real-time and high-accuracy requirements.

# B. Datasets

**ModelNet-40C.**  ModelNet-40C (Sun et al., 2022) serves as a robustness benchmark for point cloud classification, designed to evaluate the ability of architectures to handle real-world distribution shifts. It extends the original ModelNet-40 test set by introducing 15 common corruption types, grouped into three categories: transformations, noise, and density variations. These corruptions simulate practical challenges like sensor errors and LiDAR noise, offering a realistic assessment of model performance under diverse and challenging conditions.

**ShapeNet-C.**  ShapeNetCore-v2 (Chang et al., 2015) is a widely used dataset for point cloud classification, containing 51,127 3D shapes spanning 55 categories. It is partitioned into three subsets: 70% for training, 10% for validation, and 20% for testing. To evaluate the robustness of models under real-world conditions, (Mirza et al., 2023) augmented the test set with 15 types of corruptions, mirroring those in ModelNet-40C. These corruptions, created using the open-source implementation from (Sun et al., 2022), resulted in a modified version of the dataset known as ShapeNet-C.

**ScanObjectNN-C.**  ScanObjectNN (Uy et al., 2019) is a real-world dataset for point cloud classification, comprising 2,309 training samples and 581 testing samples across 15 categories. To assess robustness, (Mirza et al., 2023) applied 15 unique corruption types to the test set, using the approach described in (Sun et al., 2022). The resulting modified dataset is referred to as ScanObjectNN-C.

# C. More Experiments

**Comparison and Advantages of SMART-PC over BFTT3D in Backpropagation-Free Mode.**  Unlike BFTT3D, which uses class-based prototypes from the source dataset during adaptation, our method relies solely on the target dataset without accessing any source information. Another key distinction is efficiency: as shown in Table 4, our method in the backpropagation-free mode achieves significantly higher FPS compared to MATE, SVWA, BFTT3D, CloudFixer, and DDA. In terms of performance, our method also outperforms BFTT3D across three corrupted datasets, as reported in Table 1. Together, Table 4 and Table 1 demonstrate that our skeleton-based pretraining enables the model to learn robust

| Method | FPS |
|---|---|
| DDA | 0.04 |
| CloudFixer | 1.07 |
| BFTT3D | 6.83 |
| SVWA | 10.86 (for $N_v=2$) |
| MATE | 10.79 |
| **SMART-PC** | **59.52** |

*Table 4.* Comparison of inference speed (FPS) across different adaptation methods on the ModelNet40-C dataset.

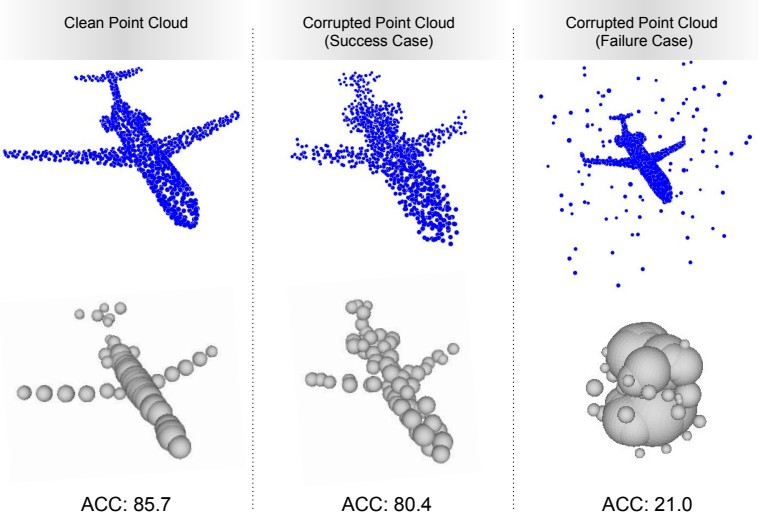

*Figure 6.* Visualization of a clean point cloud and its corrupted versions. The predicted skeleton remains stable even under corruption, supporting accurate classification (ACC: 85.7 and 80.4 in the clean and success cases, respectively). In the failure case, the corruption leads to misaligned skeletons and significantly lower performance (ACC: 21.0).

geometric features, allowing effective adaptation with only statistical parameter updates, achieving both high accuracy and fast inference.

**Analyzing the Impact of BatchNorm Statistic Updates.** Updating the running mean and variance in BatchNorm layers has been shown to improve robustness under covariate shift (Nado et al., 2020). Furthermore, the AdaBN (Li et al., 2018) paper supports the significance of this mechanism by stating that: "label related knowledge is stored in the weight matrix of each layer, whereas domain related knowledge is represented by the statistics of the Batch Normalization (BN)". This highlights that updating BN statistics is a meaningful and effective approach for handling domain shifts.

During pretraining, our method learns more abstract and robust features through the skeleton prediction branch. These features are resilient to corruption, such that during test-time adaptation, simply updating the statistical parameters of the BatchNorm layers (i.e., running mean and variance) can effectively suppress noise without requiring backpropagation.

As shown in Figure 6, our method maintains strong accuracy even under corruption. The clean input (left) yields a high classification accuracy of 85.7%, while the corrupted input still achieves 80.4% in a successful case, demonstrating robustness. In contrast, a failure case (right) leads to significant misalignment, dropping accuracy to 21.0%. These results indicate that the learned skeletal representation acts as a noise-resistant abstraction, preserving semantic structure under moderate distortions.

To further analyze the effect of BatchNorm statistics update in our setting, we visualized the impact of distribution shift on the BatchNorm input and how updating the statistics can help realign the model to the new distribution. In Figure 7, the Gaussian solid curves represent the statistics of the input data. As observed, the source distribution (blue) aligns well with the accumulated statistics from pre-training (black dashed curve), resulting in a centered and scaled distribution with zero mean and unit variance. However, when facing a distribution shift (red) in the center column, the pre-training-time accumulated statistics no longer align with the corrupted target distribution, leading to an inconsistent input distribution to the subsequent layers—compared to what the model has seen during training (i.e., covariate shift). This misalignment is clearly visible in the center column as the distance between the red solid curve and the black dashed curve. By updating the BatchNorm statistics, the running mean and variance shift toward the target distribution. This helps mitigate the covariate shift introduced by the target domain (right column), moving the statistics (green dashed curves) closer to the data distribution (solid curves).

This pattern is consistent across different channels of both the first BatchNorm layer in the encoder (top row) and the classification head (bottom row). For each BN layer, channel input values are aggregated across batch samples and tokens, and plotted as histograms of the values. Similar to the input data statistics, the BN statistics curves are plotted as Gaussian curves, computed from the corresponding running mean and running variance.

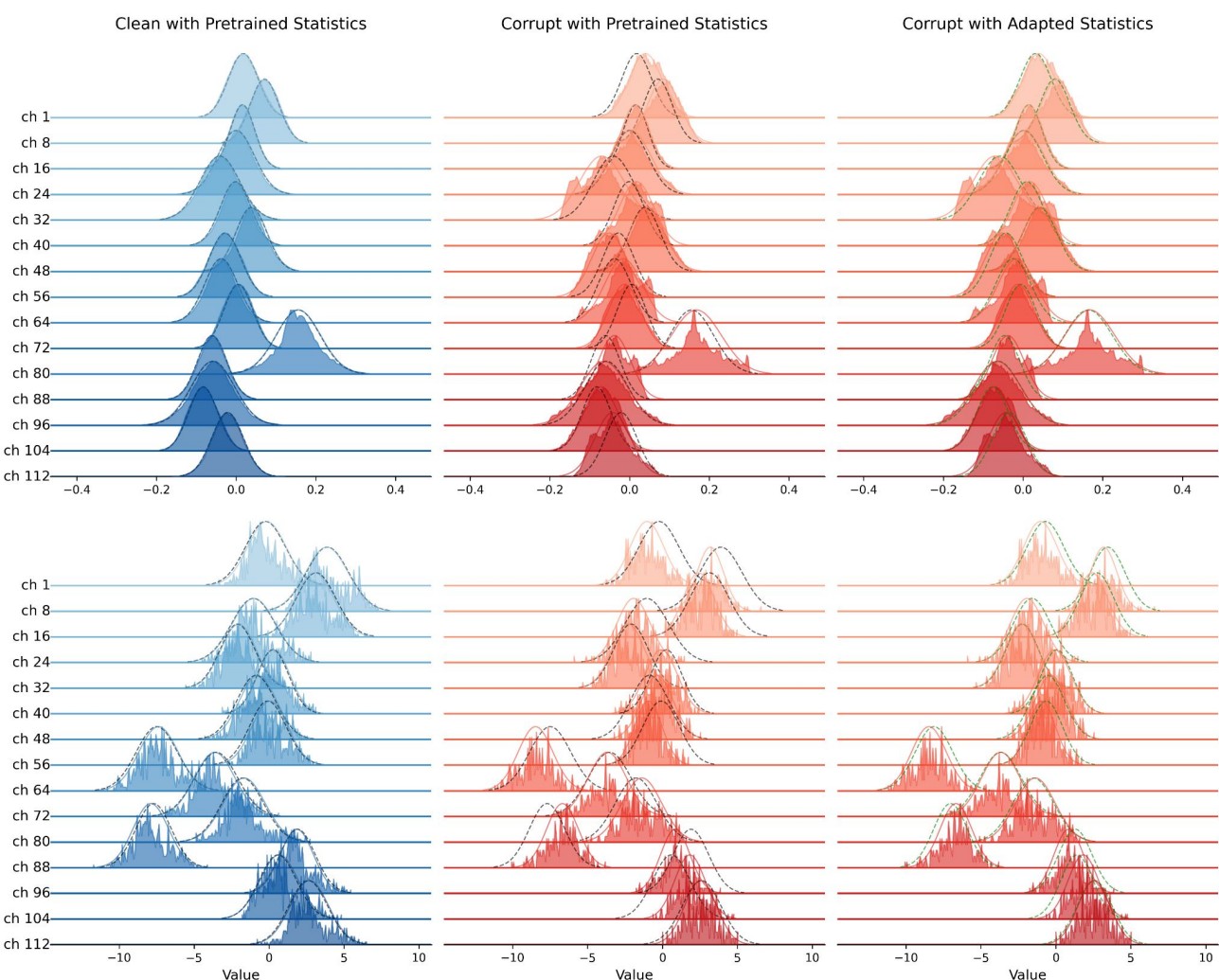

*Figure 7.* Distribution of the input (solid colors) and the running statistics (dashed curve) of BatchNorm layers across different channels. Top: first BN layer in the encoder, Bottom: first BN layer in the classifier. Blue: clean source data, Red: corrupted target data, Solid Curve: Gaussian curve of data statistics, Black Dashed Curve: pretrained accumulated statistics, Green Dashed Curve: adapted statistics.

# D. Detailed Results

In this section, the performance of our method is presented in Table 5, Table 6, and Table 7 showcasing detailed results for each corruption across all datasets. Our method consistently outperforms previous approaches in source-only (without adaptation), standard adaptation, and online adaptation modes. In most corruption scenarios, SMART-PC achieves higher accuracy compared to prior methods, demonstrating its robustness and effectiveness in handling distribution shifts across ModelNet40-C, ShapeNet-C, and ScanObjectNN-C. The improvements are particularly significant in challenging corruption types, highlighting the advantages of leveraging skeletal representations for test-time training.

# E. More Visualizations

**Skeletal Reconstruction.** In Figure 8, we present several 3D objects showcasing the original point clouds (blue dots) and their corresponding skeletal spheres. Each skeletal sphere is defined by a skeleton point (the center of the sphere) and its associated radius, which represents the local geometric structure around the skeleton point. The spheres collectively capture the essential geometric and structural features of the objects.

This visualization demonstrates the capability of our model to learn meaningful and compact representations of 3D shapes through skeletal abstraction. The skeleton effectively captures the underlying structure while filtering out high-frequency

*Table 5.* Top-1 Classification Accuracy (%) for all distribution shifts in the ModelNet40-C dataset. All results are based on the PointMAE backbone trained on a clean training set and adapted to the OOD test set with a batch size of 1. Results marked with * indicate reproduced outcomes (adaptation only) using the pretrained models from the MATE GitHub repository. Results marked with ** indicate full reproduction from scratch, including both pretraining and adaptation. The symbol "†" denotes adaptation using only BatchNorm statistical parameter updates, without backpropagation.

| Corruptions: | uni | gauss | backg | impul | upsam | rbf | rbf-inv | den-dec | dens-inc | shear | rot | cut | distort | oclsion | lidar | Avg. |
|---|---|---|---|---|---|---|---|---|---|---|---|---|---|---|---|---|
| Org-SO* | 66.6 | 59.2 | 7.2 | 31.8 | 74.6 | 67.7 | 69.8 | 59.3 | 75.1 | 74.4 | 38.0 | 53.7 | 70.0 | **38.6** | 23.4 | 54.0 |
| MATE-SO* | 59.7 | 51.3 | **28.2** | 55.3 | 71.5 | 57.4 | 60.7 | 65.2 | **77.4** | 67.1 | 30.2 | 62.3 | 61.9 | 37.2 | 19.9 | 53.7 |
| MATE-SO** | 60.8 | 53.3 | 28.8 | 53.4 | 70.9 | 57.9 | 59.6 | **68.9** | 77.1 | 67.5 | 31.7 | 65.6 | 62.0 | 33.9 | **24.6** | 54.4 |
| SMART-PC-SO | **81.8** | **79.5** | 13.6 | **65.4** | **84.3** | **75.7** | **77.8** | 62.0 | 65.9 | **73.4** | **42.7** | **69.1** | **73.8** | 36.3 | 24.4 | **61.7** |
| DUA | 65.0 | 58.5 | 14.7 | 48.5 | 68.8 | 62.8 | 63.2 | 62.1 | 66.2 | 68.8 | 46.2 | 53.8 | 64.7 | 41.2 | 36.5 | 54.7 |
| TTT-Rot | 61.3 | 58.3 | 34.5 | 48.9 | 66.7 | 63.6 | 63.9 | 59.8 | 68.6 | 55.2 | 27.3 | 54.6 | 64.0 | 40.0 | 29.1 | 53.0 |
| SHOT | 29.6 | 28.2 | 9.8 | 25.4 | 32.7 | 30.3 | 30.1 | 30.9 | 31.2 | 32.1 | 22.8 | 27.3 | 29.4 | 20.8 | 18.6 | 26.6 |
| T3A | 64.1 | 62.3 | 33.4 | 65.0 | 75.4 | 63.2 | 66.7 | 57.4 | 63.0 | 72.7 | 32.8 | 54.4 | 67.7 | 39.1 | 18.3 | 55.7 |
| TENT | 29.2 | 28.7 | 10.1 | 25.1 | 33.1 | 30.3 | 29.1 | 30.4 | 31.5 | 31.8 | 22.7 | 27.0 | 28.6 | 20.7 | 19.0 | 26.5 |
| MATE-Standard* | 69.8 | 61.8 | **18.9** | 63.9 | 72.5 | 64.0 | 66.0 | **74.0** | 80.8 | 71.0 | 36.7 | 69.2 | 66.3 | **38.4** | 29.9 | 58.9 |
| MATE-Standard** | 75.3 | 70.9 | 23.0 | 64.7 | 79.2 | 67.9 | 69.3 | 76.5 | 84.0 | 75.9 | 47.2 | 71.8 | 71.6 | 37.5 | 29.7 | 63.0 |
| SMART-PC-Standard | **82.4** | **80.1** | 12.0 | **67.1** | **84.5** | **76.0** | **78.6** | 67.3 | 72.9 | **73.3** | **43.9** | **72.6** | **73.5** | 37.4 | 24.8 | **63.1** |
| MATE-Online* | 80.6 | 79.5 | 20.7 | 71.5 | 82.6 | 78.1 | 80.7 | 78.1 | **86.6** | 79.6 | 54.9 | 78.4 | 77.4 | **45.4** | **49.6** | 69.6 |
| MATE-Online** | 82.1 | 78.3 | 29.6 | 74.1 | 84.3 | 77.4 | 79.2 | 79.4 | 86.7 | 78.8 | 55.2 | 79.1 | 76.9 | 47.8 | 49.8 | 70.6 |
| SMART-PC-Online† | 85.0 | 83.2 | 31.6 | 77.6 | **85.9** | 79.2 | 80.8 | 77.8 | 79.3 | 77.8 | 60.3 | 80.6 | **76.8** | 45.2 | 40.4 | 70.8 |
| SMART-PC-Online | **85.4** | **84.0** | **49.4** | **79.7** | 85.7 | **80.1** | **81.3** | **81.7** | 82.7 | 78.3 | **60.5** | **82.6** | 76.7 | 44.4 | 41.8 | **72.9** |

*Table 6.* Top-1 Classification Accuracy (%) for all distribution shifts in the ShapeNet-C dataset. All results are based on the PointMAE backbone trained on a clean training set and adapted to the OOD test set with a batch size of 1. Results marked with * indicate reproduced outcomes, while "†" denotes adaptation using BatchNorm statistical parameters without backpropagation.

| Corruptions: | uni | gauss | backg | impul | upsam | rbf | rbf-inv | den-dec | dens-inc | shear | rot | cut | distort | oclsion | lidar | Avg. |
|---|---|---|---|---|---|---|---|---|---|---|---|---|---|---|---|---|
| Org-SO* | 77.4 | 71.8 | 8.6 | 54.4 | 77.9 | 75.5 | 76.0 | 85.3 | 76.5 | 80.5 | 57.1 | 85.1 | 76.0 | 11.0 | 7.1 | 61.3 |
| MATE-SO* | 69.7 | 63.3 | 2.1 | 50.6 | 71.1 | 70.2 | 72.1 | **85.9** | **77.8** | 75.6 | 44.0 | **85.4** | 70.3 | 7.0 | 3.1 | 56.5 |
| SMART-PC-SO | **80.6** | **78.5** | **11.4** | **61.3** | **81.6** | **81.1** | **81.5** | 84.9 | 74.4 | **81.1** | **64.1** | 85.0 | **79.9** | **11.8** | **10.0** | **64.5** |
| DUA | 76.1 | 70.1 | 14.3 | 60.9 | 76.2 | 71.6 | 72.9 | 80.0 | 83.8 | 77.1 | 57.5 | 75.0 | 72.1 | 11.9 | 12.1 | 60.8 |
| TTT-Rot | 74.6 | 72.4 | 23.1 | 59.9 | 74.9 | 73.8 | 75.0 | 81.4 | 82.0 | 69.2 | 49.1 | 79.9 | 72.7 | 14.0 | 12.0 | 60.9 |
| SHOT | 44.8 | 42.5 | 12.1 | 37.6 | 45.0 | 43.7 | 44.2 | 48.4 | 49.4 | 45.0 | 32.6 | 46.3 | 39.1 | 6.2 | 5.9 | 36.2 |
| T3A | 70.0 | 60.5 | 6.5 | 40.7 | 67.8 | 67.2 | 68.5 | 79.5 | 79.9 | 72.7 | 42.9 | 79.1 | 66.8 | 7.7 | 5.6 | 54.4 |
| TENT | 44.5 | 42.9 | 12.4 | 38.0 | 44.6 | 43.3 | 44.3 | 48.7 | 49.4 | 45.7 | 34.8 | 48.6 | 43.0 | 10.0 | 10.9 | 37.4 |
| MATE-Standard | 77.8 | 74.7 | 4.3 | **66.2** | 78.6 | 76.3 | 75.3 | **86.1** | **86.6** | 79.2 | 56.1 | 84.1 | 76.1 | **12.3** | **13.1** | 63.1 |
| SMART-PC-Standard | **80.8** | **78.9** | 8.9 | 60.4 | **81.8** | **81.1** | **81.7** | 84.8 | 78.4 | **80.8** | **63.7** | **84.9** | **79.8** | 11.5 | 8.8 | **64.4** |
| MATE-Online* | **81.5** | 78.6 | **40.9** | **75.9** | **81.6** | 79.7 | 80.1 | **84.9** | **85.9** | 81.8 | 70.8 | **85.1** | 79.0 | **14.2** | **16.6** | **69.1** |
| SMART-PC-Online† | 80.4 | 78.7 | 21.0 | 72.7 | 80.9 | 80.9 | **80.6** | 82.5 | 78.3 | 80.9 | 70.1 | 82.5 | 79.0 | 10.5 | 9.7 | 65.9 |
| SMART-PC-Online | 81.2 | **80.5** | 28.9 | 74.3 | 81.2 | **80.7** | 80.5 | 83.1 | 81.0 | 80.4 | **73.2** | 82.8 | **79.0** | 10.0 | 10.2 | 67.1 |

*Table 7.* Top-1 Classification Accuracy (%) for all distribution shifts in the ScanObjectNN-C dataset. All results are based on the PointMAE backbone trained on a clean training set and adapted to the OOD test set with a batch size of 1. Results marked with * indicate reproduced outcomes (adaptation only) using the pretrained models from the MATE GitHub repository. Results marked with ** indicate full reproduction from scratch, including both pretraining and adaptation. The symbol "†" denotes adaptation using only BatchNorm statistical parameter updates, without backpropagation.

| Corruptions: | uni | gauss | backg | impul | upsam | rbf | rbf-inv | den-dec | dens-inc | shear | rot | cut | distort | oclsion | lidar | Avg. |
|---|---|---|---|---|---|---|---|---|---|---|---|---|---|---|---|---|
| Org-SO* | 21.7 | 18.8 | 16.9 | 18.4 | 22.2 | **46.0** | 47.0 | **72.1** | **69.4** | **48.9** | 35.6 | **73.0** | 49.4 | 6.7 | 9.3 | 37.0 |
| MATE-SO* | 20.3 | 32.2 | **18.9** | 21.2 | 20.5 | 35.6 | 36.7 | 69.9 | 66.6 | 38.9 | 28.7 | 70.4 | 39.4 | 8.3 | 9.8 | 34.5 |
| MATE-SO** | 15.3 | 20.1 | 13.4 | 11.0 | 15.3 | 28.7 | 29.4 | 69.2 | 64.5 | 32.9 | 25.0 | 70.6 | 33.4 | 9.1 | 9.0 | 29.8 |
| SMART-PC-SO | 26.7 | **37.7** | 16.9 | 21.3 | 27.2 | 44.2 | **48.9** | 69.5 | 56.3 | 48.5 | **43.2** | 72.3 | 48.0 | 8.4 | 11.0 | 38.7 |
| DUA* | 30.5 | 40.1 | 10.2 | 23.6 | 29.9 | 43.7 | 46.1 | 68.3 | 66.3 | 48.5 | 38.9 | 68.7 | 48.4 | 8.6 | 8.1 | 38.7 |
| SHOT* | 30.2 | 34.1 | 16.2 | 22.6 | 22.6 | 32.4 | 32.1 | 45.5 | 45.0 | 34.5 | 29.3 | 47.8 | 36.2 | 7.1 | 8.1 | 29.6 |
| TENT* | 29.5 | 31.6 | 17.6 | 24.8 | 27.2 | 31.0 | 32.4 | 40.7 | 35.0 | 30.2 | 26.6 | 36.6 | 29.3 | 10.5 | 12.4 | 27.7 |
| MATE-Standard* | 27.5 | 29.4 | 14.3 | **22.2** | 25.6 | 40.8 | 42.0 | **73.7** | 63.2 | 45.1 | 35.3 | 73.3 | 45.3 | 7.1 | 9.3 | 36.9 |
| MATE-Standard** | 22.9 | 31.5 | 13.4 | 17.0 | 24.3 | 34.1 | 35.5 | 69.5 | 63.7 | 39.1 | 28.4 | 68.8 | 38.7 | 7.7 | 6.2 | 33.4 |
| SMART-PC-Standard | **27.5** | **39.1** | **19.3** | 21.5 | **29.8** | 44.2 | **48.9** | 68.3 | 60.2 | **49.4** | **45.4** | 70.1 | 49.1 | 8.4 | 12.2 | 39.6 |
| MATE-Online* | 33.0 | 44.1 | 13.3 | 25.3 | 29.1 | 36.8 | 37.7 | 73.3 | 65.2 | 37.2 | 31.3 | 72.6 | 40.6 | 7.2 | 7.4 | 36.9 |
| MATE-Online** | 29.4 | 33.9 | 16.0 | 25.5 | 32.5 | 34.8 | 38.2 | 69.4 | 66.6 | 40.6 | 30.6 | 70.2 | 40.3 | 8.1 | 7.9 | 36.3 |
| SMART-PC-Online† | 39.4 | **54.6** | 19.6 | **40.1** | 40.3 | 54.4 | 55.9 | **73.3** | **69.5** | 55.1 | 50.1 | **74.9** | 57.8 | 6.9 | 7.9 | **46.7** |
| SMART-PC-Online | **42.3** | 53.7 | **23.8** | 37.5 | **41.6** | **57.0** | 57.0 | 71.1 | 68.8 | **57.3** | **53.4** | 72.1 | **58.2** | **8.4** | **8.4** | 47.4 |

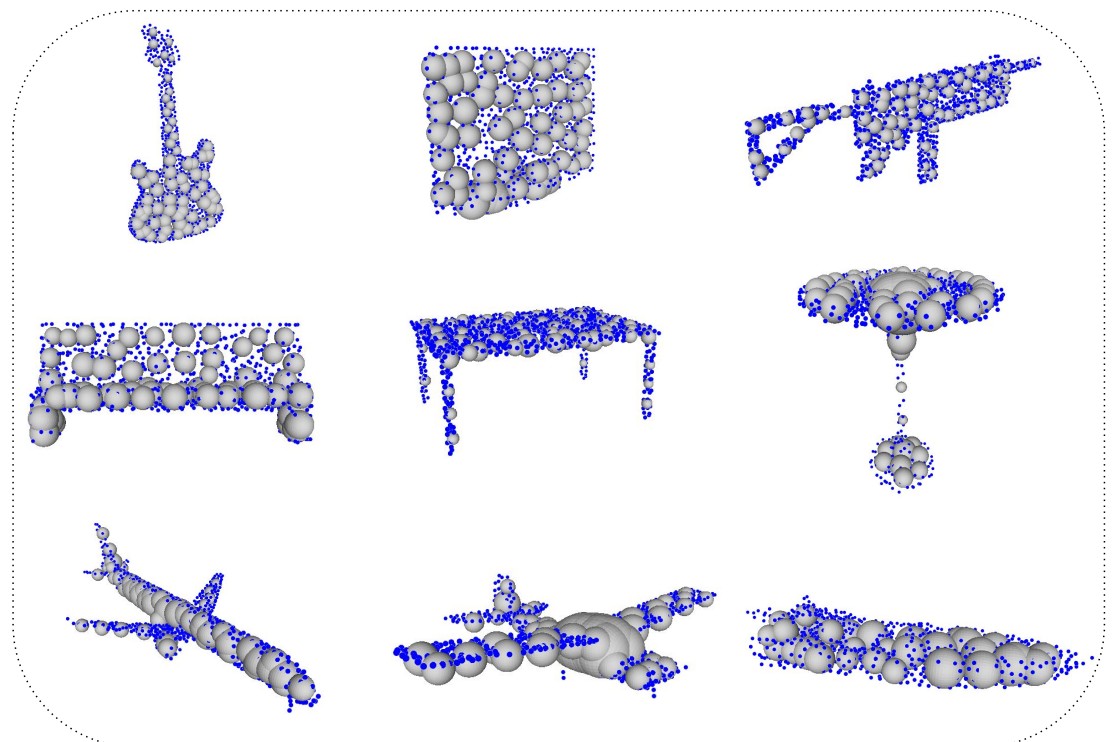

*Figure 8.* Visualization of 3D objects with original point clouds (blue dots) and their corresponding skeletal spheres.

noise, enabling the model to focus on the fundamental geometry of the objects. The diversity of shapes in this figure—ranging from guitars and airplanes to furniture—highlights the robustness of our approach in generalizing across different object classes. These visualizations also provide insight into how the skeletal abstraction simplifies complex point cloud data into manageable representations, facilitating better performance in downstream tasks such as classification and adaptation under challenging conditions. This compact representation ensures that the geometric structure is preserved, even when the original point clouds are corrupted or noisy.

