# OpenReview forum: "SMART-PC: Skeletal Model Adaptation for Robust Test-Time Training in Point Clouds"
_ICML.cc/2025/Conference — ICML 2025 poster_

### Official Review · Reviewer_pfv6 · 2025-02-19

**Overall Recommendation:** 3

**Summary:**

The paper titled introduces a novel skeleton-based framework, SMART-PC, designed to enhance the robustness and efficiency of 3D point cloud classification models during test-time training (TTT). This paper leverages skeletal representations to extract robust geometric features that are less sensitive to corruptions, enabling the model to adapt effectively to test-time distribution shifts. Extensive experiments on several benchmarks demonstrates its effectiveness

**Claims And Evidence:**

Refer to the Strengths and Weaknesses

**Essential References Not Discussed:**

Refer to the Strengths and Weaknesses

**Experimental Designs Or Analyses:**

Refer to the Strengths and Weaknesses

**Methods And Evaluation Criteria:**

Refer to the Strengths and Weaknesses

**Other Comments Or Suggestions:**

Refer to the Weakness.

**Other Strengths And Weaknesses:**

Goodness:

1.	This paper leverages skeletal representations to extract robust geometric features that are less sensitive to corruptions.

2.	Extensive experiments on several benchmarks demonstrates its effectiveness

Weakness:

1.	Why leveraging skeleton representation can eliminate the need for back propagation during adaptation is confusing, it would be better to adding more explanations or analysis.

2.	Since the authors claim the skeleton representation is more robust than current representation like point cloud, it will be better to demonstrate its generalization and  extend the skeleton representation to more methods, not limited to MATE.

3.	During the training process of predicting the skeletons, is there any more supervision object? Current framework predicts the skeleton(position+radius) of the input via an implicit way (Eq. 7,8). I am not sure whether it truly works, for the reason that just adding extra reconstruction branch can well help the model getting more inner information from the data.[1,2]

4.	The results of MATE[3] (both standard and online) on ModelNet40-C is not consistant with the paper.

5.	In Tab.1, lacking comparsion with current methods like BFTT3D[4], [5],

6.	It will be more convincing to adding more comparison with other methods about the efficiency in Fig.3.

7.	The writing needs to be improved. This point will not affect my rating score.

[1] Improving Language Understanding by Generative Pre-Training

[2] Generative Pretraining from Pixels

[3] MATE: Masked Autoencoders are Online 3D Test-Time Learners

[4] Backpropagation-free Network for 3D Test-time Adaptation

[5] Test-Time Adaptation in Point Clouds: Leveraging Sampling Variation with Weight Averaging

**Questions For Authors:**

Refer to the Weakness. I will improve my rates when the weaknesses are solved.

**Relation To Broader Scientific Literature:**

Refer to the Strengths and Weaknesses

**Theoretical Claims:**

Refer to the Strengths and Weaknesses

---

> ### Author Rebuttal · Authors · 2025-03-29
>
> We thank the reviewer for recognizing the novelty of our skeleton-based framework and the effectiveness of skeletal representations, as well as our thorough experimental validation.
>
> **Weakness**
>
> **1 and 2**. Our method supports two modes of adaptation: one with backpropagation and one that is backpropagation-free. The goal of the backpropagation-free mode is to demonstrate that, during pretraining, the use of a skeletal prediction branch encourages the model to learn **robust and meaningful geometric features**. As a result, during test-time adaptation, simply updating the **statistical parameters** of the BatchNorm layers (i.e., running mean and variance) is sufficient to enhance performance on corrupted datasets. This effect is validated by our results in **Table 1** of the main paper.
>
> To further support this claim, we conducted additional experiments using the BFTT3D[1] method, employing three different pretrained models:
>
>      1. Org-SO: A baseline model pretrained with only an encoder and classification head.
>
>      2. MATE-SO: A model pretrained using the MATE framework, which includes an encoder, a reconstruction decoder, and a classification head.
>
>      3. SMART-PC-SO: Our model, pretrained with an encoder, a skeleton-based decoder, and a classification head.
>
> Each of these models was evaluated using the same BFTT3D adaptation strategy. The results in Table 1 show that our pretrained model significantly outperforms both the baseline and MATE-pretrained models in the backpropagation-free setting across all three datasets. This provides strong evidence that the skeletal prediction branch enables the model to extract more **structure-aware and corruption-resilient features**, which allow effective test-time adaptation even without updating model weights via backpropagation.
>
> **Table 1: Mean Accuracy (%) of BFTT3D with Different Pretrained Models**
>
> | Dataset     | Org-SO | MATE-SO | SMART-PC-SO |
> |-|:-:|:-:|:-:|
> | ScanObjNN-C| 33.00| 33.22| **35.90** |
> | ModelNet40-C| 57.16| 54.71| **65.25**|
> | ShapeNet-C| 60.73| 53.07| **62.24**|
>
> [1] Backpropagation-free network for 3d test-time adaptation
> ***
>
> **3.** We confirm that our method does **not use any supervised ground truth for skeleton prediction**. Instead, the model learns to predict the skeleton structure (position and radius) in an **unsupervised manner** using the loss functions defined in Equations 11, 12, and 13. These losses serve as supervisory signals and include: (1) a point-to-sphere distance loss ensuring coverage of the input shape, (2) a skeleton-to-point loss encouraging compactness, and (3) a radius regularization term. Together, they provide strong geometric guidance, even without explicit supervision.
>
> Our goal is not to recover the point cloud itself (as in standard reconstruction branches), but to extract a more **structured and abstract representation** that captures the underlying geometry. The effectiveness of this approach is validated by the results in **Table 1 of the main paper**, which show that the features learned through the skeleton prediction branch lead to significantly better performance on corrupted datasets, especially in the backpropagation-free adaptation mode, compared to prior reconstruction-based methods like MATE.
> Additionally, we tested our pretrained model (SMART-PC-SO) with the BFTT3D method in the backpropagation-free setting, and compared it to Org-SO and MATE-SO. As shown in **Table 1**, our skeleton-pretrained model achieves higher robustness across corruptions, further supporting the generalizability and strength of the learned skeletal features.
> ***
>
> **4.** Since our implementation is largely based on the official MATE codebase, it is crucial to ensure the reproducibility of their reported results for a fair comparison. To this end, we faithfully reproduced their results using the exact code, hyperparameters, and pretrained models provided in their public GitHub repository (we witnessed a discrepancy in their results on ModelNet40-C and ScanObjectNN-C). For transparency and verification, we have also included the corresponding log files in our anonymous repository (https://anonymous.4open.science/r/SMART-PC-ICML-737C/).
> ***
>
> **5.**  We thank the reviewer for pointing this out. Please refer to our response to Reviewer **k4Xv** for the updated **Table 1**. We regret that, due to space constraints, we are unable to include the full content here.
> ***
>
> **6.** We conducted this experiment, and the results are shown in **Table 2** in our response to reviewer **k4Xv**. For DDA and CloudFixer, we report the FPS values directly from their respective papers. For all other methods, we measured the FPS ourselves, including the time required for adaptation. Importantly, the reported FPS for SMART-PC corresponds to the **backpropagation-free mode**, highlighting its efficiency under test-time adaptation without gradient updates.
> ***
>
> **7.** We will revise and improve the writing of our paper in the final version.

---

> > ### Comment · Reviewer_pfv6 · 2025-04-05
> >
> > The authors addressed my problems. I have updated my ratings.

---

> > > ### Author Response · Authors · 2025-04-08
> > >
> > > We sincerely thank the reviewer for carefully considering our responses and updating the rating. If there are any further questions or suggestions, we would be happy to address them.

---

### Official Review · Reviewer_J6vZ · 2025-03-12

**Overall Recommendation:** 3

**Summary:**

This paper proposed the method of test-time training for point cloud classification by leveraging skeletal representations. It aims to enhance the model's robustness to different distribution in test time samples. To this end, it introduced skeleton feature extraction branch besides classification branch to enhance the encoder and decoder of model.

**Claims And Evidence:**

The authors' claims are clear and straightforward, and they effectively address the limitations of previous work. For example, MATE lacks geometric understanding, rendering it less robust to surface-level corruptions. Additionally, the authors tackle the efficiency issues present in MATE. These limitations are directly addressed by the solutions proposed in this paper.

**Essential References Not Discussed:**

N/A

**Experimental Designs Or Analyses:**

Below are several suggestions to further enhance the experimental design:

1. Could the authors include visualizations that demonstrate how corrupted point clouds are restored or abstracted using skeleton points?
2. Would ablation studies on the skeletal losses help quantify their impact?
3. Is regularization necessary to prevent the radii from becoming excessively large?
4. Are there any failure cases that could be analyzed to better understand the method’s limitations?
5. Can the authors explore a backpropagation-free strategy for SMART-PC-Standard?

**Methods And Evaluation Criteria:**

Most parts of the Method section are clear and self-contained, and Figure 2 effectively illustrates the architecture and training pipeline of SMART-PC. Here are a few minor recommendations:

1. The notation for the mask predictors is inconsistent—MLP_skel (Equation 7) and MLP_s (Figure 2) are used interchangeably (similarly for MLP_radius and MLP_r). Please standardize these terms to avoid confusion.

2. At the top of Figure 2, MLP_r appears to be connected sequentially after MLP_s, which does not match the equations in the main paper. Could the authors clarify this discrepancy?

**Other Comments Or Suggestions:**

Please see above.

**Other Strengths And Weaknesses:**

Please see above.

**Questions For Authors:**

Please see above.

**Relation To Broader Scientific Literature:**

Key contributions acknowledged by both the authors and the reviewer include:

1. first work to improve test-time training of point cloud classification using skeleton representation.

2. This paper improves the performance while achieving efficient adaptation and inference pipeline.

**Theoretical Claims:**

This paper presents an empirical-results-driven method, without offering any theoretical contributions.

---

> ### Author Rebuttal · Authors · 2025-03-29
>
> We thank the reviewer for recognizing the novelty, clarity, and effectiveness of our method in improving robustness and efficiency for point cloud test-time training.
>
> **1. METHODS AND EVALUATION CRITERIA - inconsistent Notation for Mask Predictors**
>
> We'll correct the notations in the final version for clarity and consistency.
> ***
>
> **2. METHODS AND EVALUATION CRITERIA - Clarification on Decoder Structure in Figure 2**
>
> Our intention was to illustrate that each skeleton point has an associated radius, predicted separately using MLP\_s and MLP\_r. Both outputs are used to construct the final figure. For visualization purposes, they were shown connected. However, we acknowledge that this may cause confusion. In the final version, we will revise the figure to clearly separate these components and better reflect the equations in the main paper.
> ***
>
> **Theoretical Claims**
>
> By incorporating skeletal representation learning, we believe our model introduces a structural inductive bias that prioritizes shape and topology, which aligns with deep learning theory emphasizing that such biases improve generalization and robustness under distribution shifts [1].
>
> [1] Shortcut learning in deep neural networks. Nature Machine Intelligence
>
> ***
> **EXPERIMENTAL DESIGNS OR ANALYSES**
>
> 1. We have included a visualization [Figure 1] (https://anonymous.4open.science/r/SMART-PC-ICML-737C/skeleton.pdf) that illustrates how our method abstracts both clean and corrupted point clouds into compact and meaningful skeleton representations. The top row shows a point cloud under different conditions (clean, Uniform noise, Background noise), and the middle row relates to the corresponding skeletons. From this visualization, the model can estimate the skeleton under a uniform noise, preserving the overall shape of the clean data. This shows that the skeleton's abstract representation is less sensitive to the noise compared to the original points (Success Case). But under a harsh noise condition (right), the skeletons try to cover all the background outlier points, making the radius excessively large, hindering the effective representation of the main inlier points. Accuracy gain (bottom row) shows the trend; more visualizations will be provided in the final version.
> ***
>
> 2. We provided an ablation study in **Table 1** to quantify the impact of each skeletal loss component. The best performance is achieved with coefficients **(0.3, 1.0, 0.4)**—as suggested in the Point2Skeleton paper—showing a good improvement in mean corrupted accuracy (**72.95\%**) compared to other settings. This confirms that each loss term contributes to learning more robust features.
>
> ### Table 1: Ablation study of skeleton loss coefficients (ModelNet40 / ModelNet40-C, online adaptation)
>
> |Pt2Sphere|Sampling|RadiusReg|Source Acc(%)|Corrupted Acc(%)||
> |:-:|:-:|:-:|:-:|:-:|-|
> |1.0|1.0|0.0|91.3|67.82||
> |0.0|1.0|1.0|91.6|67.79||
> |1.0|0.0|1.0|91.6|67.80||
> |1.0|1.0|1.0|91.2|72.84||
> |0.3|1.0|0.4|91.3|**72.95**|coefficients from Point2Skeleton paper|
> ***
>
> 3. As described in the main paper, the **Radius Regularization Loss** (Equation 6) is designed to avoid instability caused by overly small radii, especially under noisy conditions. This loss encourages the model to learn **larger and more stable radii**, which improves the robustness of the skeletal abstraction. Although we do not observe excessively large radii, the **Point-to-Sphere** and **Sampling** losses (Equations 11 and 12) implicitly constrain radius size by preserving geometric consistency. As shown in **Table 1**, removing the regularization term leads to a drop in performance, confirming its importance.
> ***
>
> 4. In [Figure 1] (https://anonymous.4open.science/r/SMART-PC-ICML-737C/skeleton.pdf), we have included a failure case (right) to showcase a potential limitation of the skeleton estimation. From this figure, when the noise expands far beyond the main object's dimensions, the skeleton gets distracted by this out-of-distribution dimensionality change, and while it remains focused on the original points, its radii get excessively large in a way that it no longer represents the original point cloud's shape.
> ***
>
> 5. In standard mode, the batch size is 1, and the model is reset for each sample, so updating BN statistics (backpropagation-free) cannot help the model. To further investigate, we conducted an ablation study on the ScanObjectNN-C dataset in standard mode using the backpropagation-free setting. We tested batch sizes of 8, 16, and 32 with a single iteration for efficiency, as shown in **Table 2**. It shows that in standard mode, SMART-PC can improve performance on corrupted datasets in the backpropagation-free setting when the batch size is increased.
>
> ### Table 2: Ablation study of SMART-PC (standard mode and backpropagation-free) on ScanObjectNN-C.
> |Batch Size|Iteration|Mean Acc.(%)||
> |:-:|:-:|:-:|-|
> |1|--|38.7|SMART-PC-SO (source only)
> |8|1|39.73||
> |16|1|39.96||
> |32|1|39.98||

---

> > ### Comment · Reviewer_J6vZ · 2025-04-04
> >
> > The authors addressed the concerns and questions that the reviewer has. I will maintain my initial rating, "weak accept".

---

> > > ### Author Response · Authors · 2025-04-08
> > >
> > > We sincerely thank the reviewer for taking the time to read our responses and for maintaining a positive recommendation. If there are any additional concerns or suggestions, we would be happy to hear them and address them to further improve our work.

---

### Official Review · Reviewer_k4Xv · 2025-03-12

**Overall Recommendation:** 3

**Summary:**

This paper proposes a test-time adaptation framework for point cloud recognition, establishing a novel self-supervised fine-tuning paradigm that utilizes Skeletal Representation as a pretext task. By predicting skeletal points and their corresponding radii, the method extracts noise-insensitive geometric features. The authors claim that their approach eliminates the need for backpropagation during adaptation, significantly reducing computational overhead. Extensive comparative experiments are conducted, and the algorithm is validated under diverse experimental conditions.

**Claims And Evidence:**

The paper presents two main claims. The first is the use of Skeletal Representation as a pretext task to adapt the model to target domain data, which is both interesting and proven to be effective. The second claim is the direct updating of the mean and variance in the batch normalization (BN) layers to fine-tune model parameters without backpropagation, which indeed offers a novel perspective on model parameter adaptation. Both of these claims are clear and well-articulated.

**Essential References Not Discussed:**

[1] and [2], published at ECCV 2024 and CVPR 2024 respectively, are highly relevant to this study, focusing on test-time adaptation for point cloud recognition. While [1] is not cited in this work, Paper 2 is mentioned in the related work section with the statement: 'In addition to these approaches, several works on test-time adaptation have explored updating model parameters during inference to handle distribution shifts effectively.' However, [2] presents a classic test-time adaptation algorithm for point cloud recognition that does not require backpropagation, which aligns closely with the motivation of this paper. Despite this, it is neither included in the comparative experiments nor analyzed in the discussion.


[1]. Shim H, Kim C, Yang E. CloudFixer: Test-Time Adaptation for 3D Point Clouds via Diffusion-Guided Geometric Transformation[C]//European Conference on Computer Vision. Cham: Springer Nature Switzerland, 2024: 454-471.

[2]. Wang Y, Cheraghian A, Hayder Z, et al. Backpropagation-free network for 3d test-time adaptation[C]//Proceedings of the IEEE/CVF Conference on Computer Vision and Pattern Recognition. 2024: 23231-23241.

**Experimental Designs Or Analyses:**

1. The comparative methods in this paper appear outdated. It is recommended that the authors include more recent algorithms specifically designed for test-time adaptation in point cloud recognition [1][2], or adapt general test-time adaptation algorithms to the point cloud recognition task [3] for a more comprehensive comparison.

2. The experiments in this paper are quite comprehensive, encompassing both online adaptation and standard adaptation tests. The analysis of the batch normalization statistics, in particular, further substantiates the rationale for the lightweight design.


[1]. Shim H, Kim C, Yang E. CloudFixer: Test-Time Adaptation for 3D Point Clouds via Diffusion-Guided Geometric Transformation[C]//European Conference on Computer Vision. Cham: Springer Nature Switzerland, 2024: 454-471.

[2]. Wang Y, Cheraghian A, Hayder Z, et al. Backpropagation-free network for 3d test-time adaptation[C]//Proceedings of the IEEE/CVF Conference on Computer Vision and Pattern Recognition. 2024: 23231-23241.

[3]. Yuan Y, Xu B, Hou L, et al. Tea: Test-time energy adaptation[C]//Proceedings of the IEEE/CVF Conference on Computer Vision and Pattern Recognition. 2024: 23901-23911.

**Methods And Evaluation Criteria:**

The proposal of using Skeletal Representation as a pretext task to assist in fine-tuning model parameters is quite interesting. Introducing this self-supervised method into the backbone appears to be highly effective for enabling the model to adapt to target domains. I believe this contribution is well-suited for research on test-time adaptation in point cloud recognition, and it could also be extended to related fields.

**Other Comments Or Suggestions:**

I recommend that the authors provide a more detailed description of the batch normalization layer parameter updates to distinguish their approach from previous methods that directly update the BN layer weights and biases by backpropagation. This would help readers more clearly understand the authors' contributions.

**Other Strengths And Weaknesses:**

The two main contributions of this paper are highly insightful, as they adeptly apply previous research to the current task and achieve performance improvements.

**Questions For Authors:**

1. In the Classification Branch, the authors mention "adding the features from the encoder and decoder together", but I did not fully understand this description. Additionally, the main network diagram (Figure 2) does not clearly illustrate which specific features are being added or how this operation is performed. Why not directly feed the features obtained from the encoder into the classification head?

2. How exactly do the authors use the source domain label data in this paper? Is it during the TTT process? (The detail can be seen in Theoretical Claims).

3. Compared to [1], which backpropagation-free method is more effective? What are the advantages of the backpropagation-free approach proposed in this paper?

[1]. Wang Y, Cheraghian A, Hayder Z, et al. Backpropagation-free network for 3d test-time adaptation[C]//Proceedings of the IEEE/CVF Conference on Computer Vision and Pattern Recognition. 2024: 23231-23241.

**Relation To Broader Scientific Literature:**

1. [1] demonstrated that constructing an efficient pretext task in point cloud recognition can effectively enable model adaptation to target domain data. The current study further explores and proposes a novel method for designing pretext tasks specifically for test-time adaptation in point cloud recognition.

2. The analysis of batch normalization layer statistics has frequently been utilized in past studies on test-time adaptation for segmentation tasks [2][3]. However, most approaches involve constructing loss functions based on the mean and variance of BN layers to minimize the discrepancy between the source and target domains. This paper introduces a new approach by directly updating these two parameters without backpropagation. Experimental results show promising outcomes while maintaining recognition accuracy.

[1]. Mirza M J, Shin I, Lin W, et al. Mate: Masked autoencoders are online 3d test-time learners[C]//Proceedings of the IEEE/CVF International Conference on Computer Vision. 2023: 16709-16718.

[2]. Shiyu Liu, Daoqiang Zhang, Xiaoke Hao. Efficient Deformable Convolutional Prompt for Continual Test-Time Adaptation in Medical Image Segmentation. AAAI 2025.

[3]. Chen Z, Pan Y, Ye Y, et al. Each test image deserves a specific prompt: Continual test-time adaptation for 2d medical image segmentation[C]//Proceedings of the IEEE/CVF conference on computer vision and pattern recognition. 2024: 11184-11193.

**Theoretical Claims:**

In Section 3.4, when discussing the "Classification Loss", the paper mentions, "we train the network using labeled source data." Does this imply that source domain data is being used during the test-time training phase? If labels from the source domain data are indeed utilized, I believe this is inappropriate. Any modules that rely on source domain data for training should not be incorporated into the TTT (Test-time Training) process; they should be part of the source domain training phase rather than the test-time training phase [1][2]. I hope the authors can clarify the details here, as I consider this to be of significant importance.

[1]. Liang J, He R, Tan T. A comprehensive survey on test-time adaptation under distribution shifts[J]. International Journal of Computer Vision, 2025, 133(1): 31-64.

[2]. Sun Y, Wang X, Liu Z, et al. Test-time training with self-supervision for generalization under distribution shifts[C]//International conference on machine learning. PMLR, 2020: 9229-9248.

---

> ### Author Rebuttal · Authors · 2025-03-29
>
> We appreciate the reviewer’s thoughtful feedback and their recognition of our contribution to pretext task design and its connection to prior work in test-time adaptation for point cloud recognition.
>
> **1. Theoretical Claims**
>
> To clarify, **our method does not use labeled source data during the test-time training phase**. The classification loss discussed in Section 3.4 is only applied **during the pre-training phase** on the source dataset to learn the initial model parameters. During test-time training, only the target domain data without labels is used, and no source labels are required. Only the encoder and decoder are adapted at test time; the classifier is frozen (Figure 2). Furthermore, we emphasize that our method **strictly follows the standard definition of Test-Time Training (TTT)**, as established in prior works such as the MATE paper.
> We acknowledge that our wording in Section 3.4 may have caused confusion, and we will clarify this distinction in the final version of the paper.
>
> ***
> **2. Comparison with Recent Test-Time Adaptation Methods**
>
> Thank you for the suggestion. As shown in **Table 1**, we repreduced **BFTT3D[2]** and **SVWA[3]** using our settings. Since SVWA produced very low results in our standard setting (MoodelNet40-C: 5.54% and ScanObjectNN-C: 21.79%), we used BS=4, It=1, and N_v=2 for adaptation with the pretrained model available on their GitHub repository. Diffusion-based methods (**CloudFixer[1]**, **DDA[4]**) are reported separately due to slow Frame/Second (see **Table 2**). Due to time constraints, we were unable to reproduce them during the rebuttal period.
>
> [1] CloudFixer: Test-Time Adaptation for 3D Point Clouds via Diffusion-Guided Geometric Transformation
>
> [2] Backpropagation-free network for 3d test-time adaptation
>
> [3] Time Adaptation in Point Clouds: Leveraging Sampling Variation with Weight Averaging.
>
> [4] Diffusion-driven adaptation to test-time corruption.
>
> ### Table 1: Top-1 Accuracy (%) Comparation With More Recent Methods (🆕)
>
> |Method|ModelNet40-C|ScanObjectNN-C|ShapeNet-C|
> |-|:-:|:-:|:-:|
> |**Source-Only**|
> |Org|54.0|37.0|61.3|
> |MATE|53.7|34.5|56.5|
> |**SMART-PC**|**61.7**|**38.7**|**64.5**|
> |**Diffusion**|
> |🆕CloudFixer-Standard|68.0|-|-|
> |🆕CloudFixer-Online|77.2|-|-|
> |🆕DDA-Standard|68.1|-|-|
> |**Standard**|
> |🆕SVWA|57.1|37.4|50.5|
> |MATE|58.9|36.9|63.1|
> |**SMART-PC**|**63.1**|**39.6**|**64.4**|
> |**Online**|
> |🆕BFTT3D|57.2|33.0|60.7|
> |MATE|69.6|36.9|**69.1**|
> |SMART-PC†|70.8|46.7|65.9|
> |**SMART-PC**|**72.9**|**47.4**|67.1|
>
>
> ### Table 2: Frame per Second Comparison (FPS), N_v is the number of sampling variations.
> |Method|FPS|
> |-|:-:|
> |DDA|0.04|
> |CloudFixer|1.07|
> |BFTT3D|6.83|
> |SVWA|10.86 , for N_v=2|
> |MATE|10.79|
> |**SMART-PC**|**59.52**|
>
> ***
> **3. Clarification on Batch Normalization Updates**
>
> During pretraining, our method learns more abstract and robust features through the skeleton prediction branch. These features are resilient to corruption, such that during test-time adaptation, simply updating the **statistical parameters** of the BatchNorm layers (i.e., running mean and variance) can effectively suppress noise without requiring backpropagation.
> As shown in [Figure 1] (https://anonymous.4open.science/r/SMART-PC-ICML-737C/skeleton.pdf), the predicted skeleton offers a noise-resistant representation that aligns corrupted inputs with clean data. Further details will be provided in the supplementary material.
>
> ***
> **4. Clarification on Feature Fusion in the Classification Branch**
>
> The encoder and decoder outputs share the same shape **(B, N, D)**, where **B** is the batch size, **N** the number of tokens, and **D** the feature dimension. These features are combined through **element-wise addition** and passed to the classification head. This design enriches the encoder’s representation with structural information from the decoder, which captures the **skeletal geometry** of the object. As shown in **Table 2** in the main paper, this improves classification accuracy. We will clarify this mechanism and revise **Figure 2** in the final version.
>
> ***
> **5. Comparison and Advantages of the Proposed Backpropagation-Free Approach over BFTT3D**
>
> Unlike **BFTT3D**, which uses class-based prototypes from the source dataset during adaptation, our method relies solely on the target dataset without accessing any source information. Another key distinction is efficiency: as shown in **Table 2**, our method in the backpropagation-free mode achieves significantly higher **FPS** compared to **MATE**, **SVWA**, **BFTT3D**, **CloudFixer**, and **DDA**.
> In terms of performance, our method also outperforms **BFTT3D** across three corrupted datasets, as reported in **Table 1**. Together, Tables 1 and 2 demonstrate that our skeleton-based pretraining enables the model to learn robust geometric features, allowing effective adaptation with only statistical parameter updates, achieving both high accuracy and fast inference.

---

> > ### Comment · Reviewer_k4Xv · 2025-04-02
> >
> > The author has addressed my concerns clearly. As mentioned by the author, some more complex experiments may not be fully completed during the rebuttal period, but the existing experimental results already demonstrate the effectiveness of the proposed method. Furthermore, the author has provided clear responses regarding the experimental setting, and I hope the authors will provide a clearer explanation in the subsequent version.
> >
> > However, I would still like to discuss one issue with the authors. Is the direction of updating the statistical parameters in BatchNorm layers interpretable? Can the distribution of BN layer statistics obtained during forward propagation on the target domain be meaningfully linked to the characteristics of the current domain shift? The authors need not conduct additional experiments but could address this question based on existing results or their observational analysis.

---

> > > ### Author Response · Authors · 2025-04-03
> > >
> > > We sincerely thank the reviewer for their positive feedback and thoughtful comments. Updating the running mean and variance in BatchNorm layers has been shown to improve robustness under covariate shift [1]. Furthermore, the ADABN [2] paper supports the significance of this mechanism by stating that ***" label related knowledge is stored in the weight matrix of each layer, whereas domain related knowledge is represented by the statistics of the Batch Normalization (BN)"*** This highlights that updating BN statistics is a meaningful and effective approach for handling domain shifts.
> > >
> > > To analyze the effect of BatchNorm statistics update in our setting, we visualized the impact of distribution shift on the BatchNorm input and how updating the statistics can help realign the model to the new distribution. In [Figure 1] (https://anonymous.4open.science/r/SMART-PC-ICML-737C/Batch_Norm_Statistics.pdf), the Gaussian solid curves represent the statistics of the input data. As observed, the source distribution (blue) aligns well with the accumulated statistics from pre-training (black dashed curve), resulting in a centered and scaled distribution with zero mean and unit variance. However, when facing a distribution shift (red) in the center column, the pre-training-time accumulated statistics no longer align with the corrupted target distribution, leading to an inconsistent input distribution to the subsequent layers—compared to what the model has seen during training (i.e., covariate shift). This misalignment is clearly visible in the center column as the distance between the red solid curve and the black dashed curve.
> > > By updating the BatchNorm statistics, the running mean and variance shift toward the target distribution. This helps mitigate the covariate shift introduced by the target domain (right column). It is evident that updating the batch statistics moves the BatchNorm statistics (green dashed curves) closer to the data distribution (solid curves).
> > >
> > > This pattern is consistent across different channels of both the first BatchNorm layer in the encoder (top row) and the classification head (bottom row). For each BN layer, channel input values are aggregated across batch samples and tokens, and plotted as histograms of the values. Similar to the input data statistics, the BN statistics curves are plotted as Gaussian curves, computed from the corresponding running mean and running variance.
> > >
> > > If there are any additional concerns or suggestions, we would be happy to hear them and address them to further improve our work.
> > >
> > > ---
> > > [1] Nado, Zachary, Shreyas Padhy, D. Sculley, Alexander D'Amour, Balaji Lakshminarayanan, and Jasper Snoek. "Evaluating prediction-time batch normalization for robustness under covariate shift." arXiv preprint arXiv:2006.10963 (2020).
> > >
> > > [2] Li, Yanghao, Naiyan Wang, Jianping Shi, Xiaodi Hou, and Jiaying Liu. "Adaptive batch normalization for practical domain adaptation." Pattern Recognition 80 (2018): 109-117.

---

### Decision · Program_Chairs · 2025-05-01

**Decision:**

Accept (poster)

**Comment:**

All reviewers expressed a positive view of the paper and recommended acceptance. The work introduces two key contributions that are both interesting and well-supported by experimental results.

First, the use of Skeletal Representation as a pretext task for domain adaptation is novel and shown to be effective. This strategy offers a compelling approach for adapting models to target domain data in a self-supervised manner. Second, the proposed method of directly updating the mean and variance in Batch Normalization (BN) layers—without relying on backpropagation—provides a fresh and efficient perspective on model parameter adaptation. Both contributions are clearly presented and well-motivated.

The authors were responsive during the rebuttal phase, addressing experimental concerns raised by reviewers through additional experiments. While these additions significantly strengthened the paper, it is also noted that a substantial amount of clarification and refinement occurred during the rebuttal process.

Overall, the paper presents solid technical contributions and demonstrates thoughtful empirical validation. Despite the initial need for clarification, the final version reflects a promising and well-executed piece of research.